# A NEW PERSPECTIVE ON APPLYING MESOSCIENCE TO EXPLORE THE MODEL GENERALIZABILITY

## ABSTRACT

The black-box nature of machine learning (ML) models, particularly neural networks, poses a significant challenge to their broader application in engineering, especially in high-risk areas where decision-making transparency and interpretability are crucial. Understanding the generalizability of ML models remains a key topic in artificial intelligence (AI), yet a unified understanding of this issue has not been established. This study introduces the concept of compromise in competition (CIC) from mesoscience to elucidate ML model generalizability. In this work, a scale decomposition method is proposed from the perspective of training samples, and the CIC between memorizing and forgetting, refined as dominant mechanisms, is studied. Empirical studies on computer vision (CV) and natural language processing (NLP) datasets demonstrate that the CIC between memorizing and forgetting affects model generalizability significantly. Moreover, techniques like dropout and L2 regularization, traditionally used to combat overfitting, can be reinterpreted through the CIC between memorizing and forgetting. Collectively, this work proposes a new perspective to explain the generalizability of ML models, in order to provide inherent support for further applications of ML models in the field of engineering.

## 1 INTRODUCTION

ML models, particularly neural networks, have been widely implemented across various engineering fields, such as plasticity prediction Mozaffar et al. (2019), material discovery Hatakeyama-Sato et al. (2020), and fault diagnosis Qin & Zhao (2022), demonstrating robust generalizability. Nonetheless, these models are often criticized for their black-box nature, meaning their prediction processes lack transparency Hassija et al. (2024). In engineering domains where safety and prediction reliability are paramount, such as medicine Shehab et al. (2022), chemical engineering Wen et al. (2024), and autonomous driving Bachute & Subhedar (2021), the interpretability of models is critically important Zhu et al. (2022).

The interplay between model generalizability and interpretability has become a critical area of research in AI applications. In biomedical research and healthcare, particularly in cancer research, ML presents numerous opportunities, including cancer detection, diagnosis, subtype classification, treatment optimization, and the identification of novel therapeutic targets in drug discovery Elemento et al. (2021). While ML models can enhance the accuracy of cancer diagnoses, improve patient prognoses, and reduce medical costs, the challenge of explainable AI persists Elemento et al. (2021). Limited interpretability may lead to a lack of trust in these technologies among healthcare professionals Alshuhri et al. (2023).

Over the years, researchers have sought to identify key factors influencing model generalizability. The "bias-variance trade-off" Geman et al. (1992), historically viewed as a foundational principle for understanding generalizability Neal et al. (2018); Yang et al. (2020), suggests that test loss can be decomposed into bias and variance. However, bias and variance are merely outcomes on test datasets, not the underlying causes of generalizability. The "model complexity-data complexity" paradigm posits that optimal generalization is achieved when model complexity aligns with data complexity Myung (2000). Numerous studies have investigated how this relationship affects generalizability Hastie et al. (2022); Mei & Montanari (2022); Schaeffer et al. (2023), yet no unified quantitative standards for model complexity Hu et al. (2021) and data complexity Ho & Basu (2002);

Li et al. (2018a); Branchaud-Charron et al. (2019) have been established. A major challenge in unifying explanations for generalizability is the complexity of ML models and their training datasets. For instance, GPT-4, with 1.8 trillion parameters Raiaan et al. (2024), requires vast training data for robust generalization, making it difficult to describe the training process with precise mathematical or physical formulas. Thus, there is an urgent need for innovative approaches to enhance the interpretability of model prediction processes.

Recently, mesoscience (Ge et al. (2007); Li et al. (2018b)), which argues that the system complexity stems from the CIC between the two (or more) coexisting dominant mechanisms, has been proposed to cope with multilevel complexities. Instead of relying on traditional mathematical and physical formulas, mesoscience analyzes the CIC to realize the connections between system behaviors and underlying mechanisms. This approach involves performing scale decomposition, refining dominant mechanisms, and analyzing their CIC. Taking two dominant mechanisms in a system as an example, with the increasing dominance of mechanism B over mechanism A, three regimes can appear in turn: mechanism A dominates, mechanism A-mechanism B compromise, and mechanism B dominates, corresponding to different system behaviors, respectively Huang et al. (2018). The principle of mesoscience has been applied in multiple complex systems successfully, e.g., chemical engineering Li et al. (2016), life sciences Qian & Beltran (2022), geology Tordesillas et al. (2021). Guo et al. (2019) proposed a research paradigm of AI, which introduces the analytical principles of mesoscience into the design of deep learning models. This paradigm has led to the development of mesoscience-guided deep learning (MGDL), which has demonstrated remarkable improvements in terms of convergence stability and predictive accuracy Guo et al. (2024). Therefore, the application of mesoscience principles offers a promising methodological approach to explore the generalizability of ML models.

## 2 METHODOLOGY

This work employs the principles of mesoscience to elucidate the generalizability of neural networks, given their extensive applications. The research framework is depicted in Figure 1.

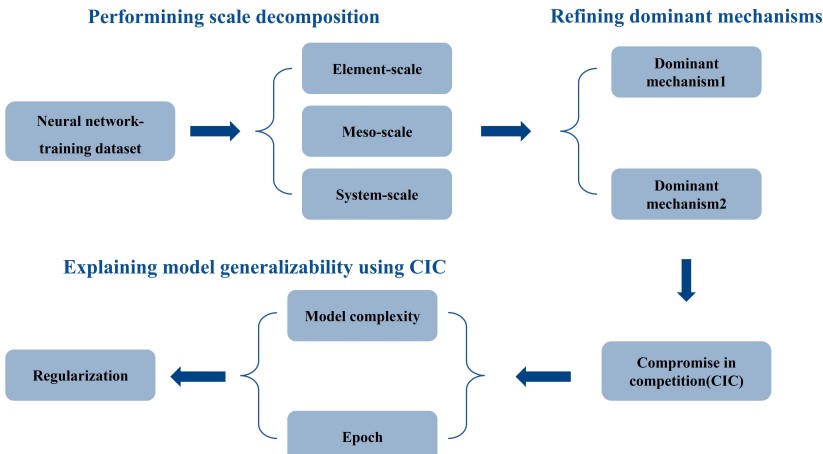

Figure 1: The research framework consists of four parts: performing scale decomposition; refining the dominant mechanisms; analyzing the CIC between them; explaining model generalizability using CIC.

For the study of complex systems, it is crucial to consider their multi-scale characteristics, particularly the quantification of meso-scale structures. A comprehensive understanding and control of system dynamics necessitate appropriate scale decomposition Ren et al. (2001), making it the initial step in mesoscience research Li & Huang (2014). This method should identify the characteristic scale that reflects the observed structure, based on the multi-scale properties of complex systems. For instance, in classical two-phase flow research, the element-scale, meso-scale, and system-scale correspond to the particle, cluster, and overall two-phase flow system, respectively, indicating that

characteristic scales must have clear physical meanings Li & Kwauk (2003). Additionally, meso-science research depends on refining dominant mechanisms. Complex systems may encompass multiple dominant mechanisms. Li & Huang (2014) proposed that it is important to group all dominant mechanisms into two integrated ones, each driving the system in opposing directions. While these mechanisms differ across systems, they adhere to the same principle of CIC. Notably, the CIC between dominant mechanisms varies across different scales Li & Huang (2014) and should be clarified. This study seeks to use CIC to uniformly explain changes in model generalizability induced by model complexity and the number of training epochs, both common factors influencing generalizability. Furthermore, the analysis of why various regularization methods mitigate overfitting will demonstrate CIC's effectiveness in explaining model generalizability.

## 2.1 EXPERIMENT SETUP

This work follows the well-established experiment setups of previous studies (Nakkiran et al. (2021); Han et al. (2020)).

- **Fully connected neural network (FCNN):** This architecture implementation is adopted from Nakkiran et al. (2021). with model complexity adjusted by modifying the width of the initial hidden layer ($w$) within the range [1, 10].

- **Four-layer convolutional neural network (Four-layer CNN):** This architecture implementation is adopted from Han et al. (2020). The models are formed by two convolutional layers and two fully connected layers. For all convolutional layers, the kernel size = 3, stride = 1, and padding = 0.

- **Five-layer convolutional neural network (Five-layer CNN):** This architecture implementation is adopted from Nakkiran et al. (2021). The models are formed by 4 convolutional stages of controlled base width [$w$, $2w$, $4w$, $8w$], for $w$ in the range of [1, 10] and one fully connected layer.

- **Text convolutional neural network (TextCNN):** This architecture implementation is adopted from Han et al. (2020). The embedding dimension is 300, and the width of the convolutional layer ($w$) is 5.

In subsequent specific experiments, FCNNs and Four-layer CNNs are used to train MNIST LeCun et al. (1998), Five-layer CNNs are used to train CIFAR-10 Krizhevsky et al. (2009), and TextCNNs are used to train TREC Li & Roth (2002).

## 2.2 SCALE DECOMPOSITION

This study introduces a scale decomposition method for complex systems, comprising neural networks and training datasets. The individual training sample, which contains the necessary features for model training, is defined as the element-scale, while the entire dataset represents the system-scale. During training, the model updates its parameters in discrete batches, as illustrated in Figure 2, highlighting the significant impact of batch size and sample composition on model generalizability. Figure 2(a) demonstrates the trade-off between increased batch size and decreased model generalizability. This finding, derived from training a Five-layer CNN ($w$ = 3) on the CIFAR-10 dataset with label noise ($p$ = 0.2), is further supported by previous work Hoffer et al. (2017), which explored the effect of batch size on model generalizability more detailedly. Additionally, this study varies the composition of samples by adjusting random seeds, as shown in Figure 2(b), where such changes significantly affect generalizability. The batch's crucial role in training dynamics and generalization qualifies it as the meso-scale. As a case study, Figure 3 illustrates the scale decomposition of neural network-CIFAR-10 dataset system.

## 2.3 DOMINANT MECHANISMS

This study systematically examines various potential mechanisms, such as bias and variance, the number of parameters and the number of training samples, the number of clean data and the number of noisy data, etc. Ultimately, memorizing and forgetting are refined as the dominant mechanisms.

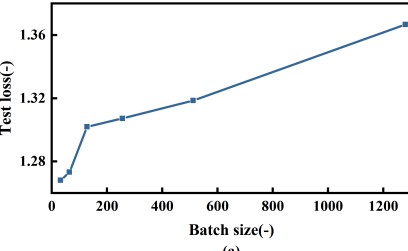 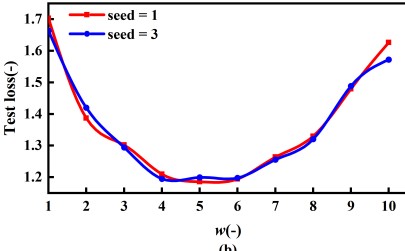

Figure 2: Training batch affects model generalizability: (a) Increasing batch size leads to a rise in test loss, suggesting reduced generalizability; (b) Variations in random seed, affecting sample composition in batches, are shown to affect the generalizability of ML models with constant architecture but varying network width ($w$).

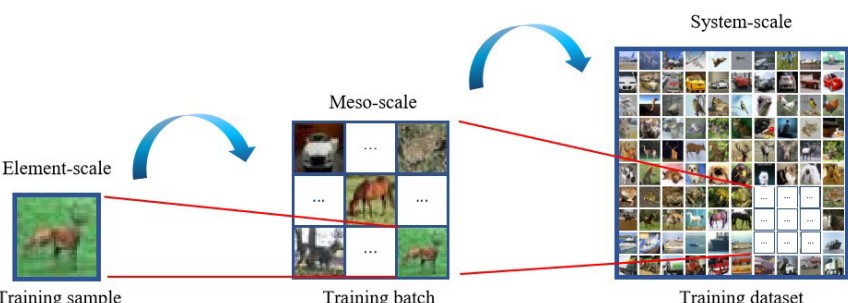

Figure 3: Taking the CIFAR-10 dataset as an example, scale decomposition is performed from the perspective of training samples. The individual training sample is the element-scale, the training batch is the meso-scale, and the entire CIFAR-10 training dataset is the system-scale.

Artificial neural networks (ANNs) discern features and formulate decisions through hierarchical abstraction, mimicking the integrative process of neuronal activity in the brain Uyanik et al. (2022). Neuroscience has long inspired advancements in AI, including CNNs LeCun et al. (1989) and reinforcement learning (RL) Mnih et al. (2015), as noted by Zador et al. (2023). Cognitive neuroscience reveals that humans continuously gather external data through sensory inputs, accumulating vast information. Like ML models, the brain processes information and makes decisions while facing risks of underfitting and overfitting. Since the brain constructs experiences and anticipates future scenarios through memorizing of learned data, insufficient memorizing may lead to underfitting. Conversely, excessive memorizing of details can result in overfitting, where reliance on past experiences hinders adaptation to new environments. Overfitting in the brain is linked to disorders such as post-traumatic stress disorder, depression, schizophrenia, and obsessive-compulsive disorder Sha et al. (2024). Gravitz (2019) demonstrates that forgetting, as an adaptive learning form, aids humans in adapting their experiences, thus preventing experiential overfitting. The importance of memorizing and forgetting extends beyond neuroscience, garnering interest from scholars in education, ecology, and linguistics Sha et al. (2024). Similar to the human brain, ML models undergo processes of memorizing and forgetting during training, significantly impacting their generalizability. For instance, scaling laws suggest that larger model sizes enhance memorization capabilities, thereby improving generalizability Kaplan et al. (2020). However, not all memorizing positively contributes to model generalizability. For example, excessive memorizing in large language models (LLMs) can lead to hallucinations Huang et al. (2023), undermining content reliability. Traditionally, forgetting is viewed as detrimental to model performance McCloskey & Cohen (1989), but recent insights from neuroscience suggest that beneficial forgetting is an adaptive function that enhances model generalizability Peng et al. (2021). Therefore, this study refines memorizing and forgetting as the dominant mechanisms, given their crucial role in influencing model generalizability.

Over time, researchers have adapted various definitions tailored to specific problems Carlini et al. (2023); Pondenkandath et al. (2018); Stern & Weinshall (2023). This study builds upon and extends the definitions proposed by Toneva et al. (2018): during training, if a model fails to accurately predict a training sample at time $t$, but succeeds at the subsequent time step $t + 1$, it indicates that the model has memorized the sample. Conversely, if the model accurately predicts a training sample at time $t$ but fails at $t + 1$, it indicates that the model has forgotten the sample. Training samples can be categorized into three distinct sets: $S_1$, which includes samples that are neither memorized nor forgotten; $S_2$, which includes samples that are only memorized and remain unforgettable subsequently (the unforgettable examples, as defined by Toneva et al. (2018)); $S_3$, which includes samples that are both memorized and forgotten at least once. These sets are mutually exclusive, and their union constitutes the entire training dataset $S$.

A list synchronized with the training epochs tracks the model's predictions for individual samples throughout training, where zeros denote inaccurate predictions and ones indicate accurate predictions, as illustrated in Figure 4. Training samples in $S_1$ are not the focus of this study, as they are neither memorized nor forgotten by the model. According to Toneva et al. (2018), samples in $S_2$ carry limited information and thus have a negligible effect on model generalizability. Experiments on datasets without label noise, such as MNIST and CIFAR-10, confirm that removing $S_2$ does not significantly impact model generalizability. In contrast, training samples in $S_3$ have a substantial effect on model generalizability Toneva et al. (2018).

Consequently, this study focuses on the memorizing and forgetting of training samples in $S_3$ by neural networks. To quantify the model's overall memorizing and forgetting of these samples, the degree of memorizing ($\mathcal{M}$) and the degree of forgetting ($\mathcal{F}$) are introduced. Specifically, $\mathcal{M}$ denotes the fraction of samples in $S_3$ that have been memorized by the end of training, while $\mathcal{F}$ denotes the fraction of those that have been forgotten in $S_3$.

$$\mathcal{M} = \frac{N_{acc=1}}{N_{acc=1} + N_{acc=0}} \tag{1}$$

$$\mathcal{F} = \frac{N_{acc=0}}{N_{acc=1} + N_{acc=0}} \tag{2}$$

where $N_{acc=1}$ denotes the number of training samples in $S_3$ predicted accurately by the end of training. $N_{acc=0}$ denotes the number of training samples in $S_3$ cannot predicted accurately by the end of training.

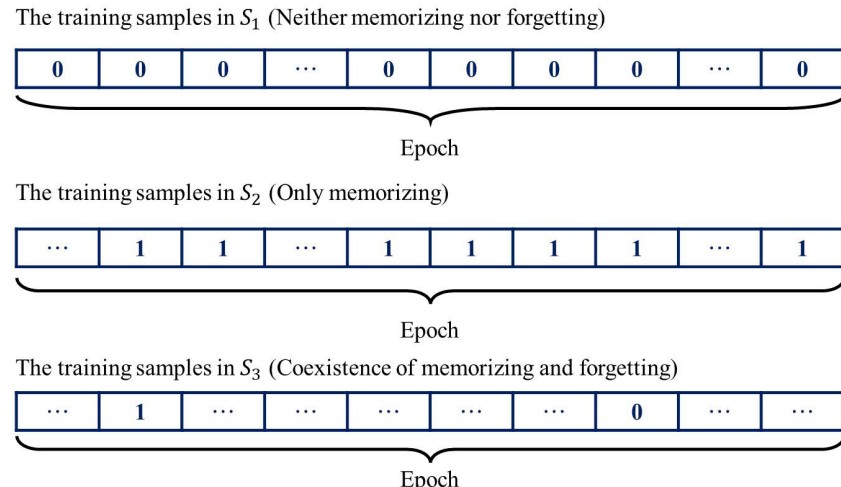

Figure 4: The model's memorizing and forgetting of samples in $S_1$, $S_2$, $S_3$ during training. $S_1$ includes the samples that are neither memorized nor forgotten; $S_2$ includes the samples that are only memorized and remain unforgettable; $S_3$ includes the samples that are both memorized and forgotten at least once, where zeros denote inaccurate predictions and ones indicate accurate predictions.

## 2.4 THE CIC BETWEEN MEMORIZING AND FORGETTING ON DIFFERENT SCALES

The CIC between dominant mechanisms exhibits variations on different scales Li & Huang (2014), and thus should be clarified. The analysis process, exemplified by training a Five-layer CNN ($w = 3$) on the CIFAR-10 dataset with label noise ($p = 0.2$), is as follows: when memorizing is dominant, $\mathcal{M}$ tends towards its maximum, whereas when forgetting is dominant, $\mathcal{F}$ tends towards its maximum. Figure 5 illustrates that on the element-scale (points A and B), the extremum tendencies of $\mathcal{M}$ and $\mathcal{F}$ can be realized only instantaneously and alternatively. Spatially, at a specific moment, such as the completion of the 106th epoch, the model's forgetting of point A is dominant, indicating the extremum tendency of $\mathcal{F}$ is realized, while the model's memorizing of point B is dominant, indicating the extremum tendency of $\mathcal{M}$ is realized. Evidently, stability conditions do not exist on the element-scale. In the meso-scale region M (batch index = 1), the extremum tendencies of $\mathcal{M}$ and $\mathcal{F}$ still cannot be realized simultaneously. However, in this region, a spatio-temporal compromise occurs between these dominant mechanisms. Over time, $\frac{\mathcal{M}}{\mathcal{F}}$ gradually converges to a constant $C_1$, with fluctuations of a certain amplitude due to the competition between memorizing and forgetting. This analysis suggests the CIC between memorizing and forgetting on the meso-scale. In the system-scale region G, the CIC is even more pronounced, with $\frac{\mathcal{M}}{\mathcal{F}}$ converging to another constant $C_2$ over time and the fluctuations diminishing.

## 3 THE CIC BETWEEN MEMORIZING AND FORGETTING EXPLAINS MODEL GENERALIZABILITY

### 3.1 CHANGES IN MODEL COMPLEXITY

In the context of training FCNNs (for $w$ in the range of [1, 10]) on the MNIST dataset with label noise ($p = 0.4$), Figure 6(a) illustrates that training loss decreases with model complexity, while the test loss decreases initially and increases subsequently, indicative of overfitting at excessive model complexity. Figure 6(b) indicates the dynamics between $\mathcal{M}$ and $\mathcal{F}$ in the neural network-training dataset system. When the model complexity is too low ($w = 1$), forgetting is dominant relative to memorizing absolutely. The fact that the model's capacity for memorizing sufficient training data is limited, and the data memorized are prone to being quickly forgotten results in poor performance on both training and test datasets, leading to underfitting. The dominance of memorizing over forgetting continuously increases with the model complexity, and the system transitions from being forgetting-dominated to being memorizing-dominated. When $w = 3$, $\frac{\mathcal{M}}{\mathcal{F}} = 1.08$, the model exhibits the best

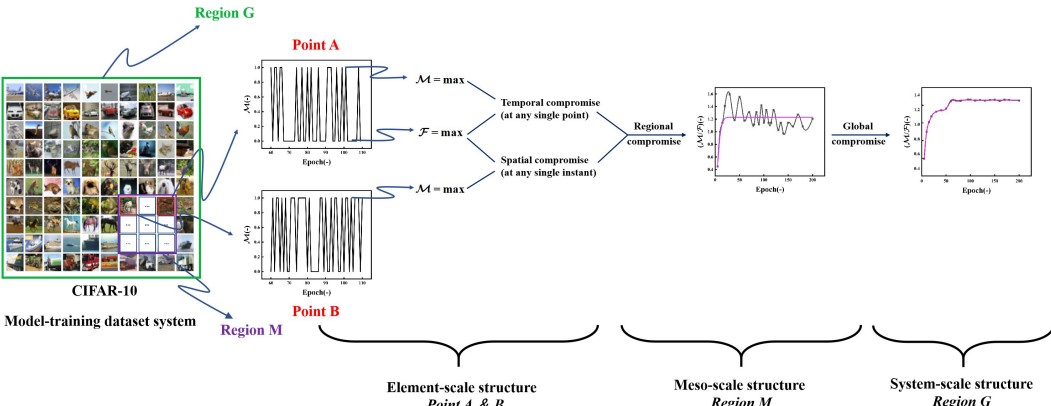

Figure 5: The CIC between memorizing and forgetting on different scales (batch index = 1). On the element scale, there is no stability conditions, while on the meso-scale and system-scale, there is CIC between memorizing and forgetting. The black line represents the change in $\mathcal{M}$ (element-scale) or $\frac{\mathcal{M}}{\mathcal{F}}$ (meso-scale and system-scale) with the number of training epochs, and the purple line represents the trend line (This analytical approach is inspired by Li et al. (2004)).

generalizability on the test dataset. On one hand, the model is capable of memorizing sufficient and accurate information. On the other hand, although the model may memorize some details (such as label noise), it forgets them eventually, thus preventing further harm to model generalizability. However, when the model complexity is too high ($w = 10$), memorizing is dominant relative to forgetting absolutely. The model has the capability to memorize a vast amount of details from the training dataset. Additionally, since the extremum tendency of $\mathcal{F}$ is inhibited, these memorized details are difficult for the model to forget, resulting in excellent performance on the training dataset but poor generalizability on the test dataset, leading to overfitting.

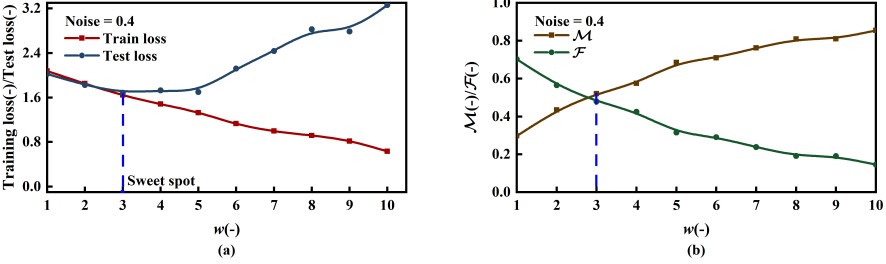

Figure 6: The change in model complexity affects the dominance of memorizing over forgetting: (a) the model tends to exhibit overfitting with the model complexity; (b) The dominance of memorizing over forgetting continuously increases with the model complexity.

The above analysis reveals that the neural network-training dataset system transitions through three distinct regimes with model complexity: When forgetting is dominant absolutely, the model performs poorly on both training and test datasets, exhibiting underfitting; When neither memorizing nor forgetting can dominate absolutely, the model shows a U-shaped test loss curve with model complexity; When memorizing is dominant absolutely, the model excels on training dataset but performs poorly on test dataset, exhibiting overfitting, as shown in Figure 7. Moreover, label noise in training datasets significantly impacts model generalizability. This study examines the effects of varying label noise on $\mathcal{M}$ and $\mathcal{F}$, as depicted in Figure 8. The findings indicate that increased label noise hinders the model's memorizing of training data, but enhances the forgetting of memorized data.

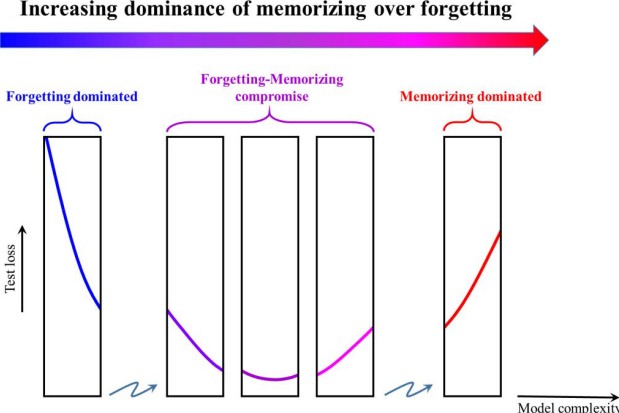

Figure 7: Three regimes occur successively with increasing dominance of memorizing over forgetting. They are the forgetting-dominated regime, the forgetting-memorizing compromising regime, and the memorizing-dominated regime, respectively.

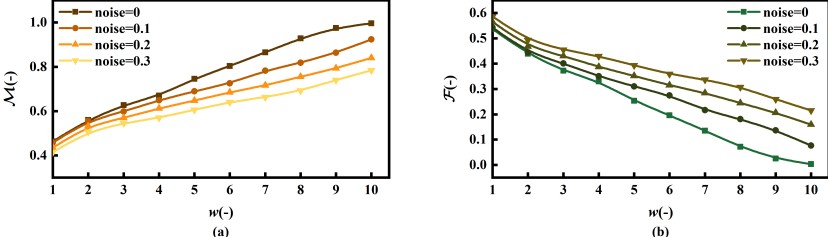

Figure 8: Increasing levels of label noise across different neural networks with different widths($w$)-CIFAR-10 dataset system consistently result in decrease in $\mathcal{M}$ and increase in $\mathcal{F}$, indicating that label noise can decrease the relative dominance of memorizing over forgetting during the training process.

## 3.2 CHANGES IN EPOCH

In the field of NLP, the training of a TextCNN with width ($w$) = 5 on the TREC dataset with label noise ($p = 0.2$) illustrates how the changing dominance of memorizing over forgetting affects model generalizability. Figure 9(a) shows that training loss decreases with epochs, while test loss initially decreases and then increases, indicating overfitting as the number of epochs becomes excessive. Figure 9(b) reveals that $\mathcal{M}$ and $\mathcal{F}$ in the neural network-training dataset system evolve continuously with epochs. With insufficient epochs, limited parameter updates lead to the absolute dominance of forgetting, resulting in inadequate memorizing and rapid forgetting, thus poor performance on both training and test datasets, indicative of underfitting. The system transitions from being forgetting-dominated to being memorizing-dominated with the number of training epochs. When the ninth epoch is completed, the model achieves optimal generalizability on the test dataset by prioritizing the fitting of regular data Arpit et al. (2017), enabling it to memorize intrinsic patterns while forgetting non-generalizable details. However, with excessive epochs, memorizing becomes absolutely dominant, causing the model to retain too many non-generalizable details. The suppression of the extremum tendency of $\mathcal{F}$ makes it difficult to forget these details, resulting in excellent training performance but poor test performance, indicative of overfitting.

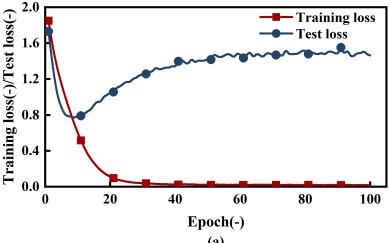 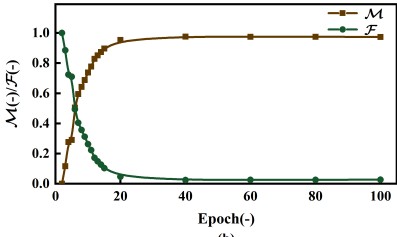

Figure 9: The changes in epoch affect the dominance of memorizing over forgetting: (a) The model tends to exhibit overfitting with the number of epochs; (b) The dominance of memorizing over forgetting continuously increases with the number of epochs.

## 3.3 REGULARIZATIONS

This study explores the effectiveness of regularization techniques, such as dropout and L2 regularization, in mitigating overfitting. In neural networks, neurons form co-adaptation relationships through interconnections and signal transmissions, capturing intrinsic patterns in training data. The dropout technique temporarily removes random neurons during training, reducing sensitivity to training data. Figure 10(a) shows that without dropout (dropout rate = 0), the model tends to overfit. An optimal dropout rate, like 0.7, enhances generalizability, whereas an excessive dropout rate, such as 0.95, can cause underfitting. Figure 10(b) indicates that without dropout, memorizing is absolutely dominant over forgetting, leading to overfitting. Moderate dropout rates, which disrupt some co-adaptations, compel the model to focus on common features of the training dataset while forgetting specific details, leading to optimal generalizability. Conversely, the excessive dropout rate increases the dominance of forgetting over memorizing which makes the model struggle to memorize but forget easily the effective information in the training dataset, leading to underfitting.

Introducing an additional penalty term, such as L2 regularization, to the loss function during training is a common method to constrain model complexity. L2 regularization reduces model complexity by adding a penalty term for the L2 norm of weight parameters to the loss function. The regularization parameter $\lambda$ is used to control the strength of the regularization term. A larger $\lambda$ increases the degree of regularization, forcing the model to adopt a simpler form and thereby reduce its complexity. For instance, in training a Four-layer CNN on MNIST with label noise (($p$=0.2)), Figure 11 illustrates that similar to dropout, L2 regularization modulates the model generalizability by controlling the relative dominance between memorizing and forgetting.

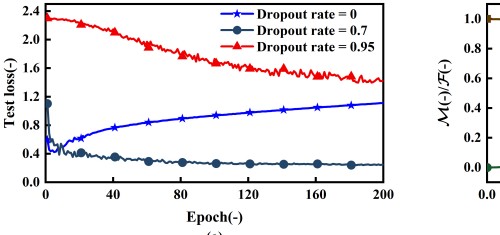 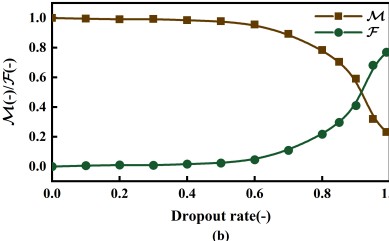

Figure 10: Dropout changes the relative dominance between memorizing and forgetting: (a) The model gradually transitions from underfitting to overfitting with dropout rate increasing from 0 to 0.95; (b) The dominance of memorizing over forgetting continuously increases with dropout rate.

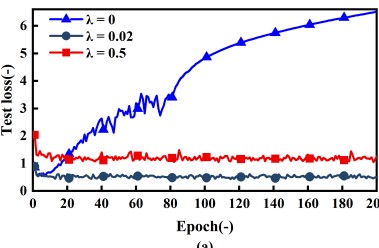 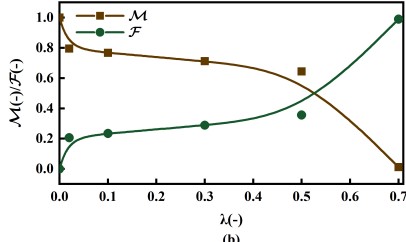

Figure 11: L2 regularization changes the relative dominance between memorizing and forgetting: (a) The model gradually transitions from underfitting to overfitting with $\lambda$ increases from 0 to 0.5; (b) The dominance of memorizing over forgetting continuously increases with $\lambda$.

## 4 CONCLUSION

This work explains the generalizability of ML models based on the principle of mesoscience, focusing on the model's memorizing and forgetting of training samples during training, and analyzes the CIC between memorizing and forgetting. Additionally, this work proposes $\mathcal{M}$ and $\mathcal{F}$ to quantify the relative dominance between memorizing and forgetting. The following conclusions are as follows:

(1) The individual training sample is considered as the element-scale, where memorizing and forgetting only compete during training; a batch of training samples is considered as the meso-scale, where memorizing and forgetting exhibit spatio-temporal compromise; and the entire training dataset is considered as the system-scale, where the spatio-temporal compromise is more evident.

(2) The increase of model complexity and the number of training epochs both can promote the extremum tendency of $\mathcal{M}$ and inhibit the extremum tendency of $\mathcal{F}$, which make the ML model-training dataset system transition from being forgetting-dominated to being memorizing-dominated gradually.

(3) Regularization methods such as dropout, L2 regularization, although proposed from different research perspectives, control the relative dominance between memorizing and forgetting to improve model generalizability essentially.

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
