# OpenReview forum: "A new perspective on applying mesoscience to explore the model generalizability"
_ICLR.cc/2025/Conference — Submitted to ICLR 2025_

### Official Review · Reviewer_ZKLt · 2024-10-25

**Soundness:** 1
**Presentation:** 1
**Contribution:** 2
**Rating:** 3
**Confidence:** 3

**Summary:**

The paper proposes analyzing model generalization (test loss) by examining subsets of training examples that are memorized or forgotten throughout the training process. By observing the ratio between M (memorized examples) and F (forgettable examples), the authors attempt to explain the behavior of the test loss. They conduct toy experiments using simplified datasets and FC and CNN models.

**Strengths:**

The idea proposed by the paper is very simple to grasp.

The "Dominant Mechanisms" section is well-written, incorporating numerous citations from other fields and providing helpful intuition and motivation.

**Weaknesses:**

**Poorly Presented:** The main weakness of this paper lies in its overly verbose and unclear presentation. Although the proposed idea, experiments, and analysis are straightforward, the authors have made the text unnecessarily wordy and complex. I have provided many suggestions below to help improve the clarity and conciseness of the writing.

**Ignoring S2:** Although the authors state their focus on the S3 subset, the size of the S2 subset is also significant (e.g., how many examples are memorized?). This aspect should be included in the analysis, as it would be valuable to understand how it impacts the results.

**Trivial Findings:** I find the results to be trivial, offering no new insights into deep learning or learning processes, as they can be explained by basic principles (e.g., variance reduction with larger sample sizes). I do not understand why the authors emphasize "mesoscience", and how looking at different scales of examples makes a difference (beyond the trivial variance reduction). If the authors wish to increase their score, they should include here (rebuttal) a clear paragraph that explicitly explains the findings, clarifies why they are not trivial, demonstrates their generalizability, and highlights their novelty compared to previous results.

**Toy Experiments and Cherry-Picking:** While it is acceptable for papers like this, which do not aim for a SOTA model or propose a new training method, to rely on toy experiments (e.g., simplified datasets, toy models, a limited number of models), the results here appear cherry-picked. It’s unclear whether the findings can be generalized, as the authors use different models to demonstrate each result. If the authors had provided a theorem on the relationships between M, F, and generalization, toy experiments might be sufficient. However, without this, I would expect robust findings, which is not the impression this paper gives.

**Questions:**

Introduction: I recommend adding two additional paragraphs—one to describe the methodology and experimental setup, and another to discuss the results and key findings.

Figure 1: Use the caption to describe the components in the figure, keeping in mind that many readers will look at the figure first to get an initial understanding of the paper before reading the text. The caption is an excellent place to briefly introduce your method and provide readers with a high-level overview of what to expect in the following sections.
At this point, I do not understand the Figure or any sentence written in the Figure. Please use the caption.

Figure 2 a: How many epochs did you use for training the models? Is it possible that the number of training steps (i.e., gradient updates) varies with each batch size? Specifically, could smaller batch sizes result in a greater number of gradient updates?

Figure 2b: The x-axis and the variable w are unclear. Please clarify these elements in the caption. The reader should not have to assume what the plot represents; it should be explained explicitly.

Line 157: The statement "mimicking the integrative process of neuronal activity in the brain" is inaccurate. If you intend to make this claim, you should support it with a citation from neuroscience research that validates it (although such a citation is unlikely to exist). You can write "inspired by..."

Suggestion for Figures: Please save the Figures in PDF format (in PowerPoint, you can print -> PDF the relevant slide, and then use a free online crop program).

Subsection 2.4: This subsection is difficult to follow. Please revise it to make your points clearer. Explicitly state what the reader should focus on and what inferences can be drawn. Currently, the subsection is too complex, and the figure caption lacks meaningful details, making it challenging for readers to follow your explanation. Additionally, you haven't defined CIC—please provide a formal (preferably mathematical) definition and follow it with a few sentences explaining this definition in simple, understandable terms.

Line 262: What do you mean by "stability condition"? Additionally, does your conclusion—“it is evident that on the element scale, there is no stability condition”—apply to every element or only to the two specific points you selected? Please ensure transparency here.

Figure 5: What is the purple line? what is the black line? Use the caption...

Subsection 2.4: The conclusion here is unclear. It appears to simply restate the well-known fact that a larger sample size results in a smaller variance for the mean statistics. Please clarify if there is an additional insight beyond this.

Figures: What does "(-)" stand for?

References (Bib): Please avoid citing published papers as "arXiv" papers. You can use resources like DBLP or Semantic Scholar to find BibTeX citations that include the appropriate journal, conference, or venue instead of referencing arXiv.

---

> ### Author Response · Authors · 2024-11-27
>
> Thank you very much for your valuable suggestions. We have revised and supplemented the manuscript accordingly. In response to your questions, our answers are as follows:
> Q1: How many epochs did you use for training the models? Is it possible that the number of training steps (i.e., gradient updates) varies with each batch size? Specifically, could smaller batch sizes result in a greater number of gradient updates?
> Reply1: The experiment involved training with six different batch sizes (32, 64, 128, 256, 512, 1280), each for 200 epchs. We controlled the number of gradient updates to remain consistent, in order to study the impact of changes in this meso-scale structures of  batches on the model generalizability.
> Q2: Line 262: What do you mean by "stability condition"? Additionally, does your conclusion—“it is evident that on the element scale, there is no stability condition”—apply to every element or only to the two specific points you selected? Please ensure transparency here.
> Reply2: Mesoscience suggests that as the scale increases, the dominant mechanisms gradually exhibit   spatio-temporal compromise. The stability condition can be used to measure the compromise in competition (CIC) between dominant mechanisms. For example, in this study, as time and scale increase, the  gradually converges to a constant, thus  can be considered a stability condition. For mesoscience research, the stability condition is crucial. “it is evident that on the element scale, there is no stability condition” apply to every element.
> Q3: Subsection 2.4: This subsection is difficult to follow. Please revise it to make your points clearer. Explicitly state what the reader should focus on and what inferences can be drawn. Currently, the subsection is too complex, and the figure caption lacks meaningful details, making it challenging for readers to follow your explanation. Additionally, you haven't defined CIC—please provide a formal (preferably mathematical) definition and follow it with a few sentences explaining this definition in simple, understandable terms.
> Reply3: This section emphasizes that the compromise in competiton (CIC)  between dominant mechanisms varies on different scales and clarifies the CIC on different scales. The CIC was introduced in the earlier introduction section.
> Q4: Figure 5: What is the purple line? what is the black line? Use the caption...
> Reply4: On the element-scale, the black line represents the change in  with the number of training epochs. On the meso-scale and system-scale, the black line represents the change in  with the number of training epochs. The purple line represents the trend line.  fluctuates around the trend line, and as the scale increases, the fluctuations gradually decrease.
> Q5: Subsection 2.4: The conclusion here is unclear. It appears to simply restate the well-known fact that a larger sample size results in a smaller variance for the mean statistics. Please clarify if there is an additional insight beyond this.
> Reply5: the  trend line (the black line in the figure) reflects the CIC between dominant mechanisms. On the element-scale, there is no compromise, only competition between dominant mechanisms. However, on the meso-scale and system-scale, due to the CIC between dominant mechanisms, the  fluctuates around the trend line (the purple line in the figure). As the scale increases, the amplitude of fluctuation decreases. This indicates a more pronounced CIC relationship between dominant mechanisms. In summary, this section clarifies the CIC between dominant mechanisms on different scales and demonstrates that  can serve as a stability condition for the complex system of neural networks and training datasets. Stability conditions play a crucial role in constructing multi-scale models of complex systems [1].
> [1] Li J, Ge W, Wang W, et al. From multiscale modeling to meso-science[M]. Berlin‐Heidelberg, Germany: Hong Kong University of Science and Technology, 2013.
> Q6: Figures: What does "(-)" stand for?
> Reply6: (-) indicates that the variable is a dimensionless quantity.
> Q7:
> Rely7: Meso-scale structures have a significant impact on system behavior. In model training, updating weights in batches rather than using individual training samples or the entire training dataset yields better results, highlighting the importance of the meso-scale. Therefore, it is necessary to study this in depth. Mesoscience systematically explores the meso-scale of different systems and has accumulated a wealth of experience. Guo et al. [2] introduced mesoscience into model design, achieving excellent generalization results. This study primarily focuses on introducing mesoscience for the first time to explain the generalizability of machine learning models. In future research, model optimization will be conducted based on this study.
> [2] Guo, L., Meng, F., Qin, P., Xia, Z., Chang, Q., Chen, J., Li, J., 2024. A Case Study Applying Mesoscience to Deep Learning. Engineering 39, 84–93.

---

### Official Review · Reviewer_w99j · 2024-11-04

**Soundness:** 1
**Presentation:** 2
**Contribution:** 2
**Rating:** 3
**Confidence:** 3

**Summary:**

The work proposes using concepts of mesoscience to explain generalization in deep learning, along with main phenomena like forgetting and memorization. In particular, it studies the (compromise in) competition of forgetting and memorization to describe how generalization happens in neural networks. Experiments are conducted using several small-scale architectural setup with prediction tasks like cifar, mnist and trec (for text).

**Strengths:**

S1: The introduction of mesoscience for explaining DL phenomena is interesting and can be promising.

S2: The authors consider a variety of architectures. Even though they are simple, they are different enough from each other and represent different setups of interest for the topic.

**Weaknesses:**

W1: As in its current state, the paper is more of an application of mesoscience to reach conclusions that are largely known in the field. While there is value in exploring the application, in terms of findings, it is hard to see the novelty of the results. For example, the tension between memorization and forgetting has been studied in depth in the past years, as is the effect of regularization and dropout.

W2: The paper does not sufficiently engage with comparing its results with previous work that falls roughly in the area of the science/physics of deep learning. Below there is some representative work in this area:

Physics of language models: Part 3.3, knowledge capacity scaling laws
Learning and generalization in overparameterized neural networks, going beyond two layers
The implicit and explicit regularization effects of dropout
Grokking: Generalization beyond overfitting on small algorithmic datasets

**Questions:**

As a suggestion for future versions of the work, showing a finding that has not been sufficiently explored in the field would be a more convincing result. In general, for any field to adopt a new method/approach there needs to be sufficient motivation from a results novelty point of view.

In addition, given that the concept of mesoscience is not familiar to the ML audience, a more comprehensive background section on the approach along with examples would make the work more approachable.

---

> ### Author Response · Authors · 2024-11-27
>
> Thank you for your valuable and helpful comments, and we have studied comments carefully and have made correction which we hope meet with approval.
> Q1. As a suggestion for future versions of the work, showing a finding that has not been sufficiently explored in the field would be a more convincing result. In general, for any field to adopt a new method/approach there needs to be sufficient motivation from a results novelty point of view.
> Reply1: In high-risk fields, such as chemical engineering and medicine, model interpretability is particularly important as it determines whether artificial intelligence can be trusted by professionals. However, widely used methods for explaining model generalizability all have their own drawbacks. For example, in specialized fields, it often requires leveraging researchers' expertise to enhance model interpretability. For example, Zhang et al. [1] incorporated doctors' expertise, BI-RADS, into the model design to improve the interpretability of the prediction process. However, this method heavily relies on the knowledge base of the researchers and can be difficult for non-experts to understand.  Besides, the “bias-variance trade-off” is historically viewed as a foundational principle for understanding generalizability. However, bias and variance are merely outcomes on test datasets, not the underlying causes of generalizability. Although the 'model complexity-data complexity' paradigm has been widely studied, there is currently a lack of a unified definition for both model complexity and data complexity. Our research not only provides a unified explanation for model generalization ability, but also refines the dominant mechanisms --forgetting and memorizing which are easily understood by the public.
> [1] B. Zhang, A. Vakanski, and M. Xian, “BI-RADS-NET-V2: A Composite Multi-Task Neural Network for Computer-Aided Diagnosis of Breast Cancer in Ultrasound Images With Semantic and Quantitative Explanations,” IEEE Access, vol. 11, pp. 79480–79494, 2023, doi: 10.1109/ACCESS.2023.3298569.

---

### Official Review · Reviewer_fa2k · 2024-11-06

**Soundness:** 2
**Presentation:** 2
**Contribution:** 2
**Rating:** 3
**Confidence:** 3

**Summary:**

This paper proposes a metric to understand model generalizability and identify memorized points. The method leverages model training dynamics, specifically tracking how predictions for examples change throughout training. The authors demonstrate this approach by adding label noise to training datasets in both computer vision and natural language processing tasks.

**Strengths:**

- This work addresses an important problem.
- Approach is simple.

**Weaknesses:**

- There is a substantial body of prior work that addresses similar questions. The authors should discuss these studies and compare at least representative methods in the experiments.
- It is unclear what new insights this paper provides regarding model training dynamics compared to existing research. The proposed method seems to overlap substantially with approaches like Maini et al. (2022), Garg and Roy (2023),  Siddiqui et al. (2022); the authors should clarify the novelty of this work in comparison.
- The experimental results are very limited, particularly given that this is an empirical study rather than a theoretical one. No deep model architectures, also, TextCNN seems like an unusual model choice for NLP experiments.

**Questions:**

- How do these observations scale with model size? Do larger models, such as BERT or even large language models (LLMs), exhibit similar patterns?
- There is extensive research exploring training dynamics in the context of spurious correlations such as Nam et al. (2020). Would it be possible to extend the experiments to examine these phenomena beyond just noisy labels?
- The authors could consider comparing against memorization measures that do not rely on training dynamics such as Feldman & Zhang (2020)
- Are the findings consistent when using the same architecture but a different optimizer? what about changes in the learning rate?

---

> ### Author Response · Authors · 2024-11-27
>
> Thank you for your valuable and helpful comments, and we have studied comments carefully and have made correction which we hope meet with approval.
> Q1. How do these observations scale with model size? Do larger models, such as BERT or even large language models (LLMs), exhibit similar patterns?
> Reply1: BERT also exhibits similar performance in the training of datasets. However, the main focus of this study is to introduce a new perspective to explain the generalizability of ML models based on mesoscience. To make this new perspective more understandable, we mainly illustrate the issue on simpler models and datasets, without using more complex models and datasets. In future research, we will extend these findings to larger models and datasets.
> Q2. There is extensive research exploring training dynamics in the context of spurious correlations such as Nam et al. (2020). Would it be possible to extend the experiments to examine these phenomena beyond just noisy labels?
> Reply2:The experiment can be further extended to training data without label noise. Figure 8 shows the change in the degree of memorizing and the degree of forgetting with the model complexity when the label noise is 0.
> Q3.The authors could consider comparing against memorization measures that do not rely on training dynamics such as Feldman & Zhang (2020)
> Reply3: Feldman & Zhang (2020) employed the leave-one-out approach to delineate a model's memorizing capacity. They proposed the view that if the removal of a sample leads to a change in the model's generalization performance, it can be considered that the model has, to some extent, memorized that sample. However, their study neglected the role of forgetting, focusing solely on memorizing. It is crucial to acknowledge that forgetting, alongside memorizing, substantially influences the model generalizability. Our study analyzes the compromise in competition(CIC) between memorizing and forgetting.
> Q4. Are the findings consistent when using the same architecture but a different optimizer? what about changes in the learning rate?
> Reply4: This is a unified explanatory method, therefore it can also be used to explore the settings of different optimizers under the same network framework. Generally, an excessively high learning rate lead to overfitting likely, which is because memorizing is absolutely dominant relative to forgetting. Conversely, an excessively low learning rate lead to underfitting likely, which is because forgetting is absolutely dominant over memorizing. Therefore, it is necessary to control the relative dominance of memorizing over forgetting within a reasonable range, which means the learning rate should be moderate.

---

### Meta-Review · Area_Chair_zfDq · 2024-12-16

**Metareview:**

After reading the reviewers comments, and reviewing the paper, we regret to recommend rejection.

The main focus of this study is to introduce a new perspective to explain the generalizability of ML models based on mesoscience, which may be promising. However, the content of the paper is relatively limited:

1.	comparison with existing (and extensive) relevant literature is lacking,
2.	experiments are very limited not taking into account several facets of the problem,
3.	it is unclear whether the results can generalize to other problems (only a few toy problems are considered), and
4.	the findings are well known and relatively trivial.

In addition, the response to the reviewers are relatively limited and do not address the concerns raised.

**Additional Comments On Reviewer Discussion:**

The reviewers raised several points, that were not fully addressed by the authors.

No ethics review raised by the reviewers, and we agree with them

---

### Decision · Program_Chairs · 2025-01-22

Reject